# Uncovering the mechanisms of synergistic drug combinations in non-small cell lung cancer through metagene-based classification

Comkrit Lomloy[1,2], Piyanut Ratphibun Yamashita[1,2], Teerasit Termsaithong[3,4], Teeraphan Laomettachit[1,4]*

1 Bioinformatics and Systems Biology Program, School of Bioresources and Technology, King Mongkut's University of Technology Thonburi, Bangkok, Thailand, 2 School of Information Technology, King Mongkut's University of Technology Thonburi, Bangkok, Thailand, 3 Learning Institute, King Mongkut's University of Technology Thonburi, Bangkok, Thailand, 4 Theoretical and Computational Physics Group, Center of Excellence in Theoretical and Computational Science, King Mongkut's University of Technology Thonburi, Bangkok, Thailand

* teeraphan.lao@kmutt.ac.th

## Abstract

Drug resistance remains a significant challenge in treating non-small cell lung cancer (NSCLC). Identifying synergistic drug combinations that simultaneously target multiple signaling pathways is crucial to overcoming drug resistance, yet challenging due to the extensive search space. To address this issue, we developed a computational framework that combines network analysis and clustering based on matrix factorization to gain mechanistic insights into highly synergistic drug combinations in the A549 NSCLC cell line. First, we used a Random Walk with Restart (RWR) algorithm to propagate the effects of drug combinations on a molecular interaction network tailored to A549 NSCLC. This approach transformed sparse drug-target data into comprehensive molecular profiles for 607 drug combinations. These profiles were then analyzed using Graph-regularized Non-negative Matrix Factorization (GNMF) to classify drug combinations based on metagenes, representing common patterns of impact on key biological pathways. Our analysis successfully identified clusters highly enriched with synergistic drug pairs. Notably, a single feature, Metagene 2, consistently drove synergy in seven of these clusters. Pathway enrichment analysis indicated that Metagene 2 is primarily associated with the interconnected RAS, MAPK, and PI3K/AKT signaling pathways. This observation led to specific mechanistic hypotheses: for instance, synergy with dasatinib appears to result from co-targeting SRC compensatory pathways, while the enhanced effects of paclitaxel combinations arise from partner drugs disrupting the PI3K/AKT pathway, which in turn modulates Tau protein activity. In conclusion, the metagene-based classification provides an interpretable and rational approach for uncovering the systemic biological mechanisms responsible for drug synergy. This framework offers a valuable tool

**Data availability statement:** All relevant data are within the manuscript and its Supporting information files. The programming scripts and related data are available from our GitHub repository: https://github.com/systemsbiomedicine/A549-drug-synergy-with-network-propagation.

**Funding:** This work was supported by King Mongkut's University of Technology Thonburi (KMUTT), Thailand Science Research and Innovation (TSRI), and National Science, Research and Innovation Fund (NSRF) (Fiscal year 2025, Grant number FRB680074/0164 to TL and TT). CL was supported by the Petchra Pra Jom Klao Master's Degree Research Scholarship from King Mongkut's University of Technology Thonburi. The funders had no role in study design, data collection and analysis, decision to publish, or preparation of the manuscript.

**Competing interests:** The authors have declared that no competing interests exist.

for designing effective, multi-target therapeutic strategies to overcome drug resistance in NSCLC.

## Introduction

Lung cancer remains a major global health issue. In 2022, an estimated 2.5 million new cases were diagnosed, accounting for 12.4% of all new cancer cases worldwide. It was also the leading cause of cancer-related deaths, with approximately 1.8 million fatalities [1]. Non-small cell lung cancer (NSCLC) accounts for approximately 85% of these cases. Among NSCLC, adenocarcinoma is the most common type, comprising around 40% of cases. NSCLC patients often face significant treatment challenges, including severe side effects and limited clinical responses due to drug resistance. Resistance to treatments is driven by various mechanisms, such as tumor heterogeneity, hypoxia, altered drug transport, and dysregulated signaling pathways [2,3]. In addition, cancer cells can develop adaptive, non-genetic resistance, involving the dynamic rewiring of cell signaling networks to bypass drug effectiveness [4,5].

Combination therapy using two or more drugs to target multiple biological pathways is a promising strategy for overcoming drug resistance, as drug combinations can provide greater efficacy than monotherapy with less systemic toxicity [6,7]. Effective examples include pairing WEE1 and CHK1/2 inhibitors [8] to accumulate DNA damage and sensitize cells to DNA-damaging agents [9], combining CDK4/6 inhibitors with endocrine therapy for metastatic breast cancer [10], and combining dabrafenib and trametinib for metastatic NSCLC with $BRAF^{V600E}$ mutation [11]. However, identifying effective drug combinations remains difficult due to the exponential number of potential drug pairs, and synergistic combinations are rare [9].

Researchers have recently explored computational approaches to accelerate the discovery of drug combinations. Machine learning algorithms have proven useful in predicting drug synergy. A prominent example is DeepSynergy [12], which utilizes drug structures and genomic information from cancer cell lines for drug synergy prediction and has become a benchmark for later models. However, these "black-box" models often lack biological interpretability regarding the mechanisms driving synergy. To improve interpretability, various machine learning methods have been developed to interpret drug synergy predictions by identifying key features like proteins or pathways that contribute to observed synergistic effects, linking prediction with mechanisms [13–18].

Alternatively, mechanistic modeling explicitly simulates biological processes of drug action through mathematical representations of cellular pathways and networks, enabling the study of drug resistance and the prediction of combination therapy effectiveness in cancer [19–23]. However, mechanistic models are complex to construct for large-scale networks.

Network analysis provides a complementary strategy by leveraging functional relationships within biological networks [24]. Cheng and colleagues (2019) [25] successfully utilized the topological structure of the human protein-protein interactome to predict drug synergy. By analyzing the network proximity between drug targets and

disease protein modules, the authors found that effective drug combinations target different but complementary neighborhoods within the disease module. This geometry offers a rational basis for designing synergistic multi-drug, multi-target strategies. Network propagation further enhances this analysis by inferring gene function based on network proximity. The technique relies on the principle that proteins/genes in close proximity are more likely to be functionally related and contribute to similar biological processes [26]. Hofree and colleagues (2013) [27] used Random Walk with Restart (RWR) to propagate cancer mutation signals of individual patients across the gene network. By combining this with factorization techniques, they identified patient subgroups based on pathway disruption, improving predictions of survival and treatment response. Moreover, Park and colleagues (2015) [28] used the RWR algorithm to propagate signals from drug targets on a protein-protein interaction network to predict pharmacodynamic drug interactions and aid drug safety and development.

This study uses the RWR algorithm on drug pair targets to propagate the signals across a curated, A549 NSCLC-specific molecular interaction network. This process converts sparse drug-target data into dense, network-wide molecular profiles that reflect both direct and indirect effects of drugs. These profiles are then analyzed using Graph-regularized Non-negative Matrix Factorization (GNMF) and hierarchical clustering to group the 607 tested drug combinations into biologically meaningful clusters. This integration helps identify drug combinations with synergistic effects on NSCLC and, importantly, facilitates the mechanistic interpretation by linking clusters of synergistic pairs to specific non-drug-target genes and key pathways, offering insights into the combination's therapeutic actions.

## Materials and methods

### Ethics statement and data source

The data used in this study were obtained from the publicly available DrugComb database (https://drugcomb.fimm.fi) [29]. This research is a purely computational analysis of in vitro cell line data. Therefore, the requirements for ethical approval from an Institutional Review Board (IRB) or informed consent are not applicable to this study.

### Method overview

A brief overview of our methods is presented in Fig 1. The process begins by **(1)** retrieving drug combination data for the A549 cell line from the DrugComb database [29]. Key information includes the names of the drugs in the drug pairs and their corresponding synergistic scores. Next, protein targets for each drug in the combinations were obtained from DrugBank [30]. **(2)** Subsequently, a cancer-specific sub-network was extracted from the Parsimonious Composite Network (PCNet) [31]. **(3)** For each drug combination, drug targets for each drug in the combination were mapped onto the extracted cancer network. **(4)** The mapped profile for each drug combination underwent network propagation via a Random Walk with Restart (RWR) algorithm to create a propagated profile. **(5)** Based on these propagated profiles, drug combinations were grouped into clusters using Graph-regularized Non-negative Matrix Factorization (GNMF)-based clustering, which categorized drug combinations by their impact on "metagenes," a group of genes participating in the related biological pathways within the cancer network. **(6)** Finally, the clusters were analyzed to determine the metagenes that associate with the drug synergy. These metagenes were used to propose the therapeutic actions of drug combinations in A549. The programming scripts and related data are available from our GitHub repository: https://github.com/systemsbiomedicine/A549-drug-synergy-with-network-propagation. The algorithms for network propagation and GNMF were derived from the Stratipy library, available at https://github.com/eraldop/stratipy/tree/master. Next, we provide details of each step in our methods.

### Data retrieval: Drug combination dataset and drug target identification

Approximately 85% of lung cancer cases are classified as non-small cell lung cancer (NSCLC); therefore, this study focuses on drug combination experiments in NSCLC, particularly the A549 cell line, which is one of the most commonly used human NSCLC cell lines in basic research and drug discovery. Drug combination data were retrieved from

**Fig 1. Overview of the methods.** (1) Data retrieval: drug combination information for the A549 cell line and drug target data are gathered. (2) Cancer network construction: a cancer-related network is built from the Parsimonious Composite Network (PCNet). (3) Mapping drug targets onto the cancer network: the targets of each drug in the combinations are mapped onto the network. (4) Network propagation: a Random Walk with Restart (RWR) algorithm is used to generate a propagated profile. (5) GNMF and unsupervised clustering: the propagated profiles of all drug combinations are analyzed with Graph-regularized Non-negative Matrix Factorization (GNMF) and clustering. (6) Identifying the therapeutic actions of drug combinations: the resulting clusters are examined to determine the mechanistic actions of the drug combinations.

DrugComb, version 1.5 [29]. After filtering for only the drug combinations tested on A549, we identified a total of 5343 drug combinations comprising 140 unique drugs. For each combination, the database provides four synergy scores that quantify the synergistic effect between drugs. The HSA synergy score compares the combined effect to that of the more effective single agent. The ZIP synergy score calculates the expected effect under the assumption that the drugs do not potentiate each other. The Bliss synergy score evaluates the combined effect, assuming the drugs act independently. Finally, the Loewe synergy score estimates the effect assuming the two drugs behave as if a single drug was combined with itself. Synergy scores may be positive or negative. A positive score indicates a synergistic effect, while a negative score denotes an antagonistic effect [32].

We gathered drug target gene information from the DrugBank database version 5.1.10 [30]. Each of the 140 unique drugs in our drug combination dataset was matched to a DrugBank ID. After matching, 126 of the 140 drugs remained, and drug combinations containing unmatched drugs were removed. Next, gene/protein target IDs associated with the remaining drugs were collected from DrugBank and converted to gene symbols to align with the node names in the cancer network (cancer network construction will be presented below). We selected only drugs with inhibitory effects on target genes/proteins, and only drug combinations containing these drugs were retained. Finally, drug combinations containing drugs whose target genes/proteins are not present in the cancer network were excluded. These steps resulted in a final dataset of 607 drug combinations (S1 File) involving 45 unique drugs (S2 File), which was used in the subsequent analyses.

**Cancer network construction**

We retrieved the Parsimonious Composite Network (PCNet) version 1.0, which is sourced from molecular network resources utilized in human disease research [31]. PCNet outperforms any individual network among a compendium of 21 evaluated molecular networks in recovering literature-based human disease-associated gene sets. It is constructed by integrating all molecular interactions that appear in at least two of the 21 networks, resulting in a comprehensive network comprising 19,781 nodes and 2,724,724 interactions. To make the network more specific to cancer, a sub-network was extracted using four cancer-related gene sets to filter the network: (1) genes from hallmark cancer pathways [33], (2) tumor suppressor and oncogenes [34], (3) recurrently mutated cancer genes discovered from cancer cell lines (Sanger UK) [35], and (4) genes from the Cancer Gene Census (COSMIC v81) [36]. A list integrating the four gene sets was compiled, resulting in a total of 2331 cancer genes. PCNet was then filtered to contain only genes from the combined cancer gene list and the edges between those genes. Any genes that do not have edges connecting to other genes were removed. The resulting cancer network consisted of 2297 genes and 204,826 interactions.

PCNet represents genes and their interactions involved in various types of human cells. To tailor this network to NSCLC, the mutation status of genes in the A549 cell line was integrated. Mutation data were obtained from the Cell Model Passports database (version 2.9.0) [37]. A total of 891 mutated genes were found, 380 of which corresponded to nodes in the cancer network. Among these, 197 exhibited a loss of function, 123 showed a gain of function, and 60 were categorized as unknowns. Genes with loss-of-function mutations and their interactions were removed from the network, as they are considered inactive and unable to transmit signals. After this adjustment, the final cancer network used for analysis included 2100 genes and 162,211 interactions (S3 File).

**Mapping drug targets onto the cancer network and network propagation**

For each drug combination, we implemented a Random Walk with Restart (RWR) algorithm based on Eq. 1.

$$x_T = \alpha \left[ I - (1 - \alpha)AD^{-1} \right]^{-1} \cdot x_0$$

(Eq. 1)

where $x_0$ is the initial state or seed vector, represented as a 2100-dimensional column vector that corresponds to the nodes in the cancer network. Nodes that are not targeted by either drug in the pair are assigned a value of 0. For the target gene nodes of the first drug, values are set to 0.5 divided by $n_1$, where $n_1$ is the total number of target genes for that drug. Similarly, the values for target gene nodes of the second drug are set to 0.5 divided by $n_2$, where $n_2$ is the total number of target genes for the second drug. Gene nodes targeted by both drugs in the pair are assigned values of $0.5/n_1 + 0.5/n_2$. The sum of the column vector $x_0$ is always equal to 1. $A$ is the adjacency matrix of the network with a dimension of $2100 \times 2100$, and $D$ is the diagonal matrix of the out-degrees of the nodes. Thus, $AD^{-1}$ is the normalized adjacency matrix. $I$ is an identity matrix and $\alpha$ is the restart probability [38]. The calculation returns $x_T$, which is a propagated profile of each drug combination after the propagation of information.

We chose a balanced value of $\alpha$ at 0.5 to ensure that information signals are sufficiently propagated beyond the immediate local neighborhood without becoming excessively diluted and uninformative throughout the broader network. The propagated profiles of the 607 drug pairs form a large data matrix $X$ with a dimension of $2100 \times 607$, where each column represents a specific drug pair and each of the 2100 rows corresponds to a node in the cancer network. This step converted sparse drug-target data into network-wide molecular profiles for 607 drug combinations.

### GNMF of the propagated profiles

GNMF was applied to factorize the propagated profile matrix $X$ of size $N \times M$, where $N = 2100$ is the number of genes in the network and $M = 607$ is the number of drug pairs in our dataset, into two non-negative matrices, $W$ and $H$, such that $X \approx W \cdot H$. Matrix $W$ has a size of $N \times K$, with each of the $K$ columns defining a metagene (a low-dimensional representation of the original genes) [39]. The entry $w_{nk}$ represents the coefficient of gene $n$ in metagene $k$. In GNMF, the metagene composition is constrained by the topology of the cancer network, with a regularization parameter of $0 \leq \lambda \leq 1$ [27]. We chose $\lambda = 0.9$ to ensure that genes closely connected in the network have similar weights in representing the same metagene, thereby preserving biological significance.

Matrix $H$ is of size $K \times M$, with each of the $M$ columns representing the propagated pattern of the corresponding drug pairs in terms of metagenes; the entry $h_{km}$ indicates the propagated score of metagene $k$ in drug pair $m$. To determine the optimal number of metagenes ($K$), we analyzed the reconstruction error as $K$ was varied from 1 to 60 and identified the "elbow point" in the error curve. This point represents the best balance between model complexity and accuracy. Once the optimal $K$ is found, we employed the GNMF with this optimal $K$ and the regularization parameter ($\lambda$) of 0.9.

### Unsupervised clustering and identifying the therapeutic actions of drug combinations

After factorization, matrix $H$ was used to cluster 607 drug combinations using Ward's hierarchical clustering method. Then, the clusters were systematically examined by computing the mean synergy score (HSA, ZIP, Bliss, and Loewe) for drug combinations within each group. The aim was to identify clusters with an average synergy score above the third quartile (Q3) of that synergy measure, highlighting a significant enrichment of highly synergistic combinations within each cluster. The metagene profiles from matrix $H$ of the identified clusters were analyzed to gain insights into the mechanisms of drug combinations in the A549 cell line.

## Results

### Propagated scores can differentiate between drug combinations with low and high synergy scores

After performing network propagation using the RWR algorithm with the selected restart probability ($\alpha$) set to 0.5, propagated profiles of 607 drug pairs across 2100 genes were obtained (S4 File). To determine if the network propagation scores of individual genes could effectively differentiate between different levels of drug synergy, we divided the drug combination synergy scores into four quarters based on the quartile values. For example, the HSA scores of the 607 drug

combinations in our dataset fall within these ranges: −16.007 to −2.375, −2.299 to 0.378, 0.382 to 3.702, and 3.707 to 29.711, corresponding to quarters 1–4, respectively. We used the same approach for the ZIP, Bliss, and Loewe scores.

We then conducted a statistical analysis using a two-sided Wilcoxon rank-sum test to compare the propagated scores of drug combinations with synergy scores in the first (bottom 25%) and fourth (top 25%) quarters for each of the 2100 genes in the network. The Wilcoxon test was chosen because the distribution of the network propagation scores is highly skewed. The p-values from the tests were adjusted using the False Discovery Rate (FDR) method to control for multiple comparisons. Genes with an adjusted p-value of less than $1 \times 10^{-4}$ were considered having a statistically significant difference in propagated scores between the first- and the fourth-quarter groups. This results in 79, 2, 257, and 844 significant genes for the HSA, ZIP, Bliss, and Loewe scores, respectively. As examples, Fig 2 displays box plots comparing the distributions of propagated scores for genes with the lowest p-values from the four different synergy scores. Each plot shows the distribution of propagated scores for the gene across the four quarters, which represent increasing levels of drug synergy. The box plot for TRAF1 shows a trend where the

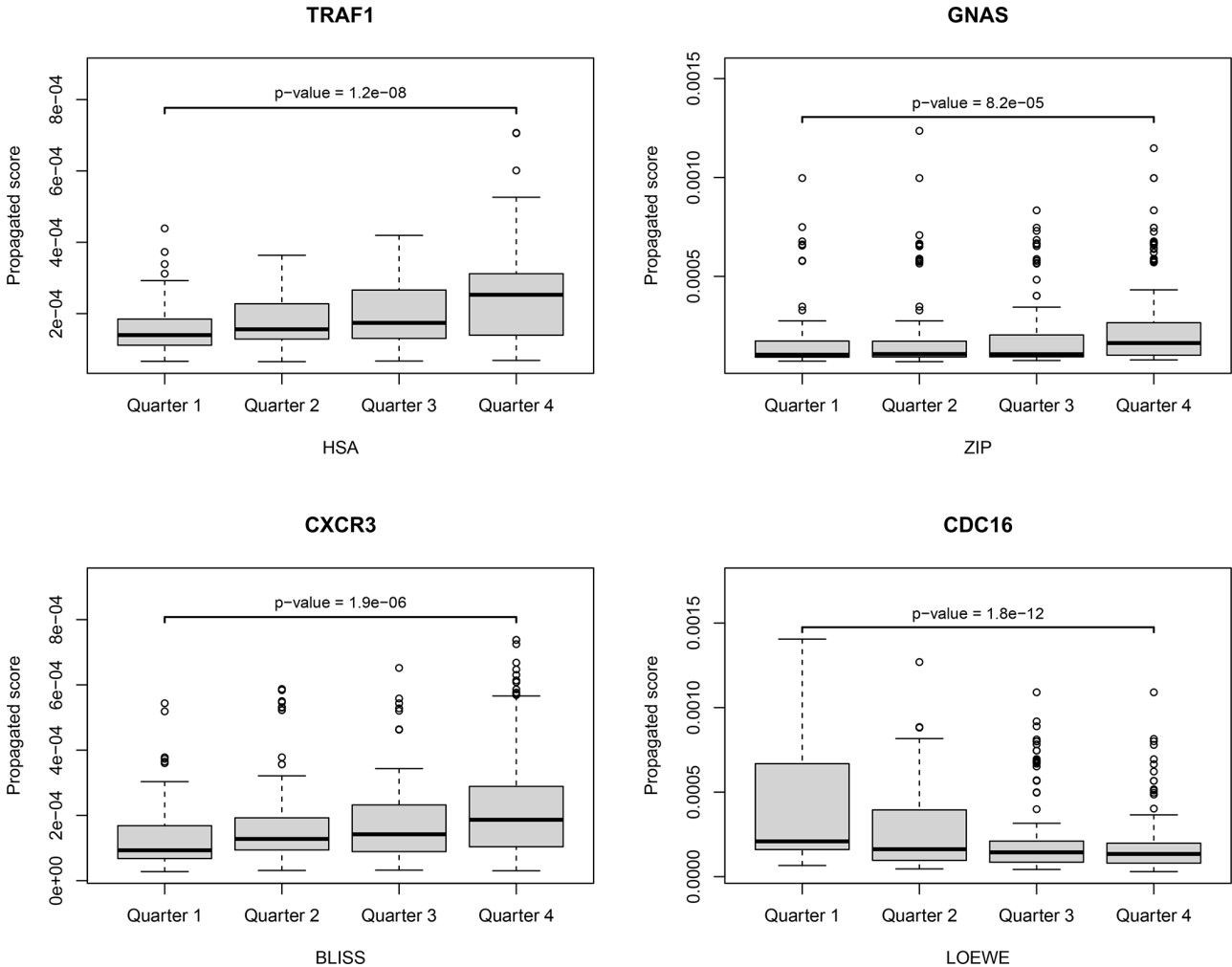

**Fig 2. Box plots comparing the distribution of propagated scores for selected genes (TRAF1, GNAS, CXCR3, and CDC16) across the four quarters of drug combination synergy scores (HSA, ZIP, Bliss, and Loewe).** The statistical significance between Quarter 1 and Quarter 4 is shown using a two-sided Wilcoxon rank-sum test with FDR correction.

propagated score rises from the first quarter to the fourth quarter. Similarly, for CXCR3 and GNAS, an increase in the median propagated score from the first quarter to the fourth quarter can be observed, although the difference is less noticeable for GNAS. In contrast, the CDC16 box plot indicates a significant decrease in the median propagated score from the first to the fourth quarter.

Additionally, the scatterplots in Fig 3 illustrate the relationship between the gene's propagated score and the synergy score for all drug combinations, where each point represents a distinct drug combination. For TRAF1, GNAS, and CXCR3, an increase in the propagated score correlates with an increase in the synergy score, with Spearman's correlation

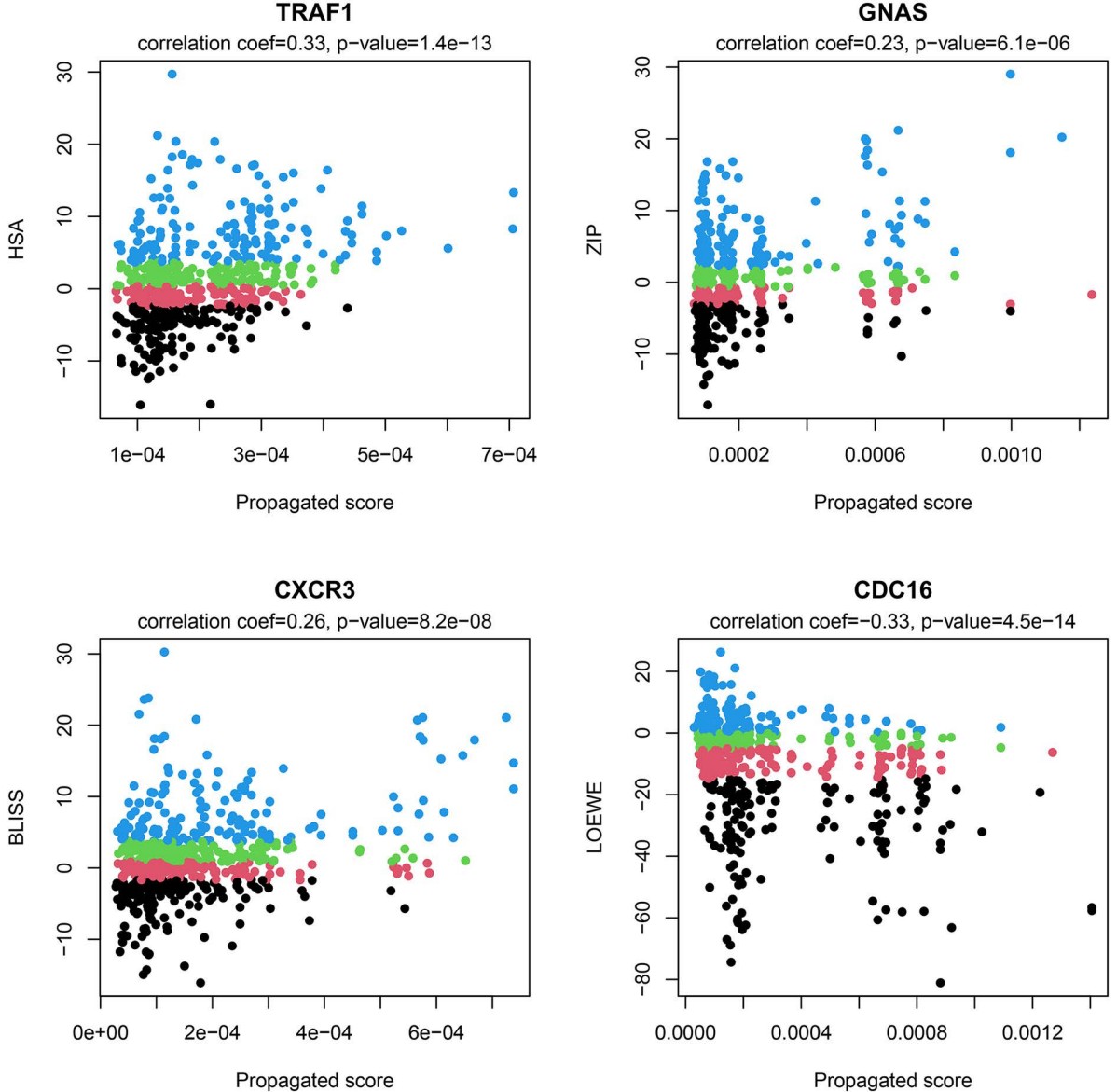

**Fig 3. Scatterplots showing the relationship between the propagated scores of selected genes (TRAF1, GNAS, CXCR3, and CDC16) and the drug combination synergy scores (HSA, ZIP, Bliss, and Loewe).** Data points are colored by quartile of their synergy score: Quarter 1 (black), Quarter 2 (pink), Quarter 3 (green), and Quarter 4 (blue). Spearman correlation coefficients and FDR-adjusted p-values are displayed above each panel.

coefficients of 0.33, 0.23, and 0.26, respectively. Conversely, for CDC16, an increase in the propagated score correlates with a decrease in the synergy score, with a Spearman's correlation coefficient of −0.33. The results of this analysis provide evidence that propagation scores of certain genes can serve as indicators of drug synergy, potentially distinguishing between high-synergy and low-synergy drug combinations.

## GNMF represents the propagated profiles of 607 drug combinations with 26 metagenes

Building on the finding that the propagated scores of many genes relate to the degree of drug synergy, we applied graph-regularized non-negative matrix factorization to the propagated profiles to reduce the high dimensionality of 2100 genes and facilitate biological interpretation. The data distribution of the propagated profiles exhibited a heavily right-skewed distribution, with most values concentrated near zero and a long tail extending toward higher values. Thus, we applied a logarithmic transformation to reduce the skewness by compressing the long tail and stretching values near 0, resulting in a distribution that is closer to normal. This adjustment minimizes the impact of extreme values (primarily corresponding to direct drug target genes/proteins) and, therefore, enhances the performance of subsequent analyses (e.g., clustering analysis).

The matrix $-\log(X)$, where $X$ is the propagated profiles, was used as input for GNMF, since the algorithm requires a non-negative matrix. A regularization parameter $\lambda$ value of 0.9 was selected to factorize the matrix while prioritizing adherence of genes in the metagenes to the cancer network structure. To determine the optimal number of metagenes ($K$), the GNMF algorithm was run with varying values of $K$, ranging from 1 to 60, while tracking the reconstruction error. The KneeLocator function identified the elbow point at an optimal $K$ value of 26. This process produced two matrices, $W$ and $H$, where $-\log(X) \approx W \cdot H$. Matrix $W$ has dimensions of 2100×26, with each row representing a gene and each column representing one of the 26 metagenes. Matrix $H$ has dimensions of 26×607, with each row representing a metagene and each column representing one of the 607 drug combinations.

## Clustering drug combinations with metagene profiles reveals clusters strongly enriched with highly synergistic combinations

The matrix $H$ represents the profiles of 26 metagenes for each drug combination, providing a compressed, biologically interpretable signature. We performed hierarchical clustering on the drug combinations using matrix $H$. We used Euclidean distance as the dissimilarity measure and Ward's D2 method for linkage, which minimizes the total within-cluster variance. The clustering results were analyzed by systematically cutting the dendrogram at every height that created a new cluster partition. For each resulting set of clusters, we calculated the average synergy score of the drug combinations within each cluster. We specifically targeted clusters with an average synergy score above the third quartile (Q3) of their respective synergy score type, indicating a high enrichment of strongly synergistic combinations. Clusters with fewer than 10 members were excluded from this analysis. From this process, we identified 16 unique clusters (S1 Table in S1 Text), some of which were nested partitions created using different cut heights. To eliminate redundancy and highlight the most important clusters, we selected only the cluster for each lineage with the highest overall synergy score. This resulted in a final set of 10 clusters, detailed in Table 1.

To determine if a specific pattern exists among these highly synergistic clusters, we created a box plot of the metagene profiles (matrix $H$) for all combinations from the ten clusters, as shown in Fig 4. The plot reveals a strong, consistent signal in Metagene 2 across most combinations. Statistical analysis showed that the signal for Metagene 2 is significantly higher than that of all other metagenes (one-sided Wilcoxon signed-rank test, FDR-adjusted p-value < 0.05). Visualizing the metagene profiles of individual clusters (Fig 5) showed that seven out of ten clusters consistently exhibited an active Metagene 2 profile, implicating this metagene as a key part of the synergy mechanism. These seven clusters feature drugs such as dasatinib, paclitaxel, quinacrine, erlotinib, gefitinib, crizotinib, pazopanib, and ruxolitinib.

**Table 1. Ten identified clusters with an average synergy score above the third quartile (Q3).**

| Cut height ($h$) | Cluster ID | Number of drug pairs in the cluster | HSA | ZIP | Bliss | Loewe | Drug with the highest frequency | Number of drug pairs containing the most frequent drug |
|---|---|---|---|---|---|---|---|---|
| 148.48 | 26 | 17 | 4.31±5.59 | | | | Crizotinib | 13 |
| 115.69 | 28 | 11 | 7.88±5.11 | 3.38±6.75 | 8.24±5.9 | | Dasatinib | 11 |
| 104.12 | 27 | 12 | 5.14±4.1 | | 4.64±4.42 | | Erlotinib | 4 |
| 102.69 | 34 | 10 | | | | 0.01±2.9 | Gefitinib | 5 |
| 66.86 | 16 | 10 | | 2.58±6.58 | 5.38±9.54 | | Mitotane | 10 |
| 133.28 | 19 | 11 | 8.37±6.48 | 7.02±7.39 | 7.74±6.48 | | Paclitaxel | 10 |
| 155.46 | 22 | 18 | 5.11±5.92 | 2.6±6.81 | 5.78±5.71 | | Pazopanib | 18 |
| 124.14 | 2 | 15 | 11.83±7.65 | 9.71±8.41 | 12±7.78 | 4.83±14.34 | Quinacrine | 15 |
| 78.19 | 69 | 10 | 3.93±7.06 | | | | Ruxolitinib | 10 |
| 190.5 | 20 | 14 | | 3.92±11.32 | 3.84±12.59 | | Vemurafenib | 11 |

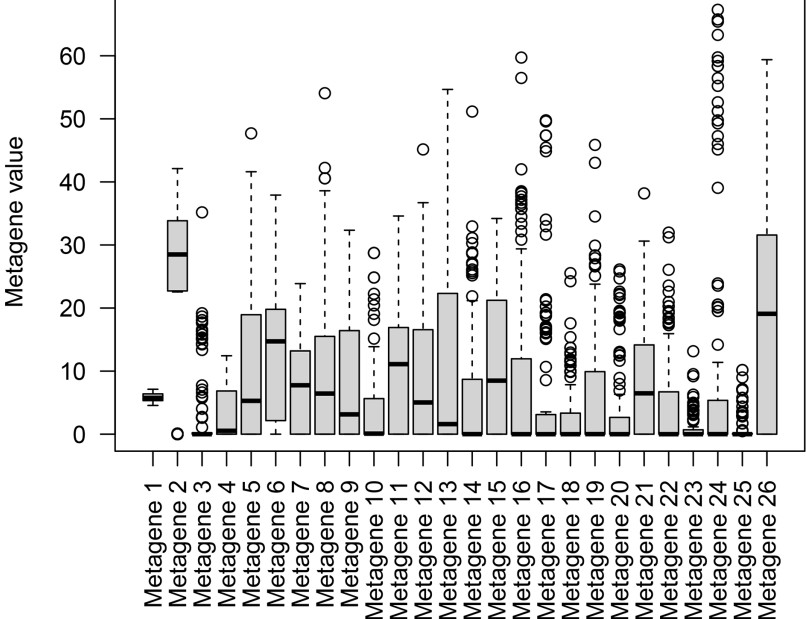

**Fig 4. Box plot of 26 metagene activation levels (from matrix $H$) across all drug combinations from the 10 highly synergistic clusters.** Metagene 2 exhibits significantly higher activation levels than other metagenes (one-sided Wilcoxon signed-rank test, FDR-adjusted p-value < 0.05).

Further evidence of Metagene 2's significance was demonstrated by hierarchical clustering all dasatinib-containing drug combinations into two groups using matrix $H$. Fig 6a illustrates that Metagene 2 distinctly separates the two groups: a low Metagene 2 profile group (Group 1) and a high Metagene 2 profile group (Group 2). As shown in Fig 6b and S2 Table in S1 Text, the high Metagene 2 group (Group 2) has significantly higher average synergy scores across the three metrics (HSA, ZIP, and Bliss) compared to the low Metagene 2 group (Group 1). The analysis suggests that although both groups contain dasatinib-related combinations, their underlying biological mechanisms, as indicated by Metagene 2, are distinct, and Metagene 2 may explain how synergy is achieved with dasatinib.

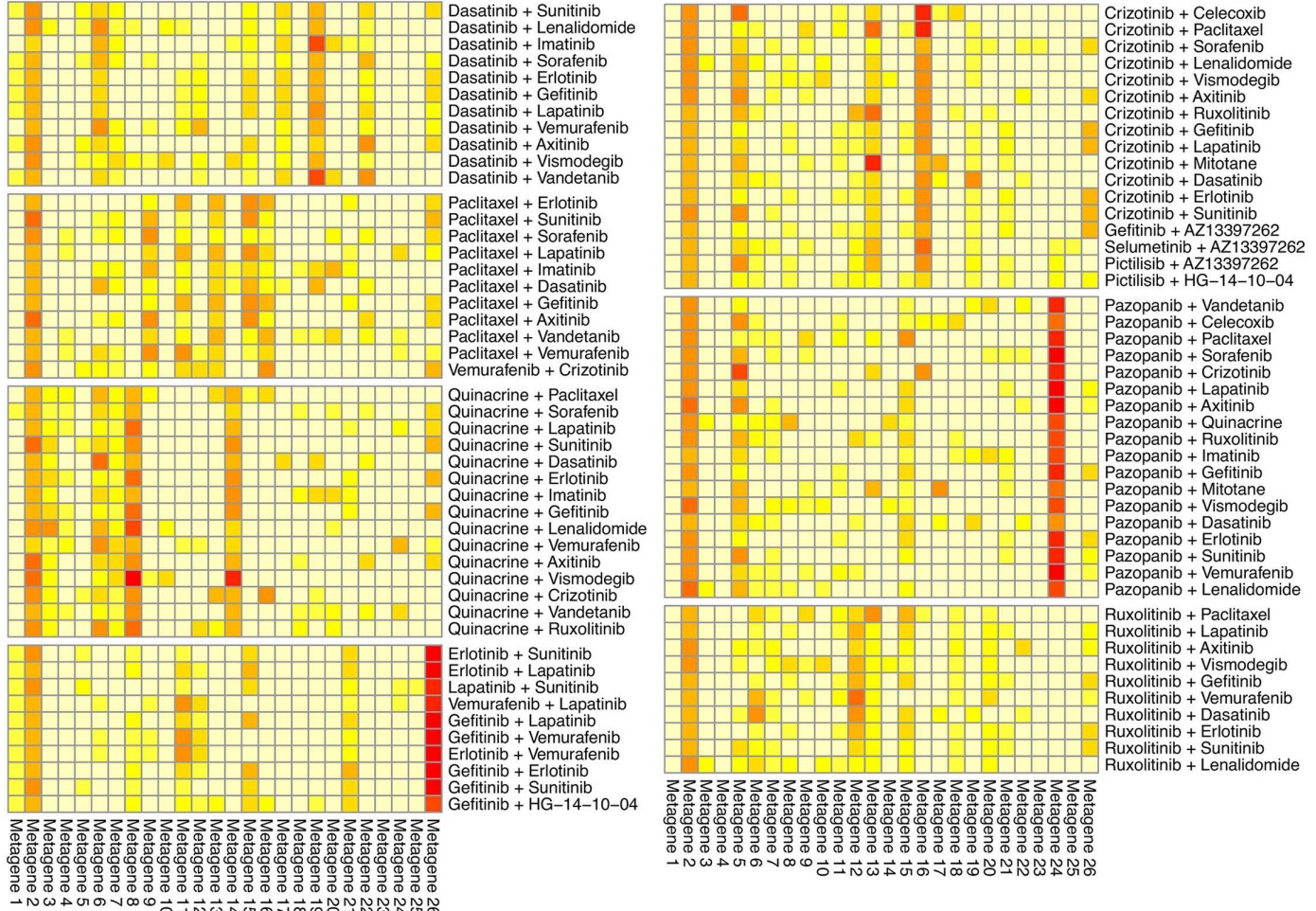

**Fig 5. Metagene profiles (matrix *H*) of seven highly synergistic clusters showing high and consistent levels of Metagene 2.**

A similar pattern is observed with paclitaxel, quinacrine, crizotinib, and pazopanib, where Metagene 2 can be used to distinguish drug combinations involving these drugs into two groups (S2-S5 Figs in S1 Text). The high Metagene 2 group shows an average synergy score significantly higher than the low Metagene 2 group (S3-S6 Tables in S1 Text). The effect is less pronounced with ruxolitinib, erlotinib, and gefitinib, where Metagene 2 does not effectively differentiate between high and low synergistic groups (S6-S8 Figs and S7-S9 Tables in S1 Text).

### Strong Metagene 2 signals of highly synergistic combinations are achieved by the interactions between the drug pairs rather than individual drugs

The seven highly synergistic clusters that share a strong Metagene 2 signal feature combinations of key drugs (dasatinib, paclitaxel, quinacrine, and others) with a group of frequently appearing partner drugs, including lapatinib, vemurafenib, sunitinib, axitinib, vismodegib, sorafenib, lenalidomide, vandetanib, and imatinib. For instance, lapatinib is present in all seven clusters. This pattern raises the question of whether the general properties of these partner drugs drive the observed synergy and the strong Metagene 2 signal. We investigated this possibility and found that, generally, combinations containing these partner drugs did not exhibit synergy scores significantly higher than zero. The only exception was

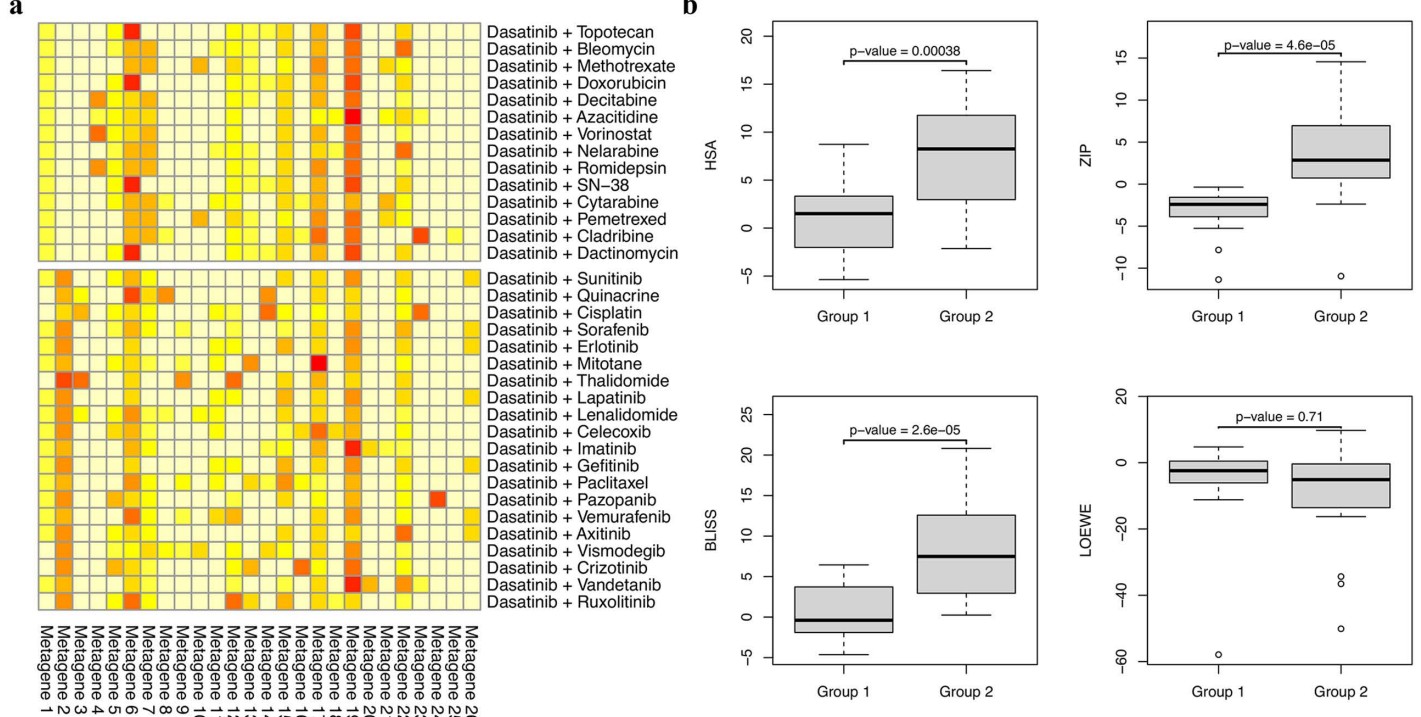

**Fig 6. Metagene profiles and synergy scores for drug combinations containing dasatinib. (a)** Metagene profiles (matrix *H*) of all drug combinations containing dasatinib. The 34 drug combinations are divided into two groups using hierarchical clustering. Group 1 includes 14 combinations with low Metagene 2 signals, while Group 2 includes 20 combinations with high Metagene 2 signals. **(b)** Box plots showing that drug combinations from Group 2 (high Metagene 2 signal) have significantly higher synergistic scores in HSA, ZIP, and Bliss compared to those from Group 1 (low Metagene 2 signal). P-values were calculated using a one-sided t-test.

vemurafenib, which showed significant synergy under the Bliss model (Fig 7a). Therefore, the high synergy seen in the seven clusters is unlikely to be driven by the inherent synergistic potential of this drug group.

Additionally, we determined that a strong signal in Metagene 2 is also not an inherent property of the partner drugs. For example, as shown in Fig 7b and 7c, many drug pairs containing lapatinib or imatinib exhibit low Metagene 2 activity. As a result, the high synergy scores and the strong Metagene 2 signal observed in the identified clusters come from specific pharmacological interactions within the drug combinations, rather than a general characteristic of any single drug.

### Genes from Metagene 2 are enriched in the PI3K/AKT, MAPK, and RAS signaling pathways

Our analysis above suggested that Metagene 2 might underlie the synergy of many drug combinations in our A549 dataset. Therefore, we analyzed the top 200 genes that contribute most to Metagene 2 (from Matrix *W*) using KEGG enrichment analysis [40], performed in the STRING database [41]. (Note that we performed the GNMF analysis with the negative values of log(propagated scores); therefore, genes that contribute the most to the metagenes have the lowest weights in Matrix *W*.) The analysis strongly implicated the PI3K/AKT, MAPK, and RAS signaling pathways (Fig 8), indicating that the coordinated regulation of these pathways governs the synergistic mechanism of these drug combinations. S10 Table in S1 Text lists the 200 genes.

The diagram in Fig 9 depicts the potential synergistic actions of drug combinations that exert coordinated influence on survival and proliferation pathways, collectively captured by Metagene 2 (outlined in a blue box): the PI3K/AKT and RAS/

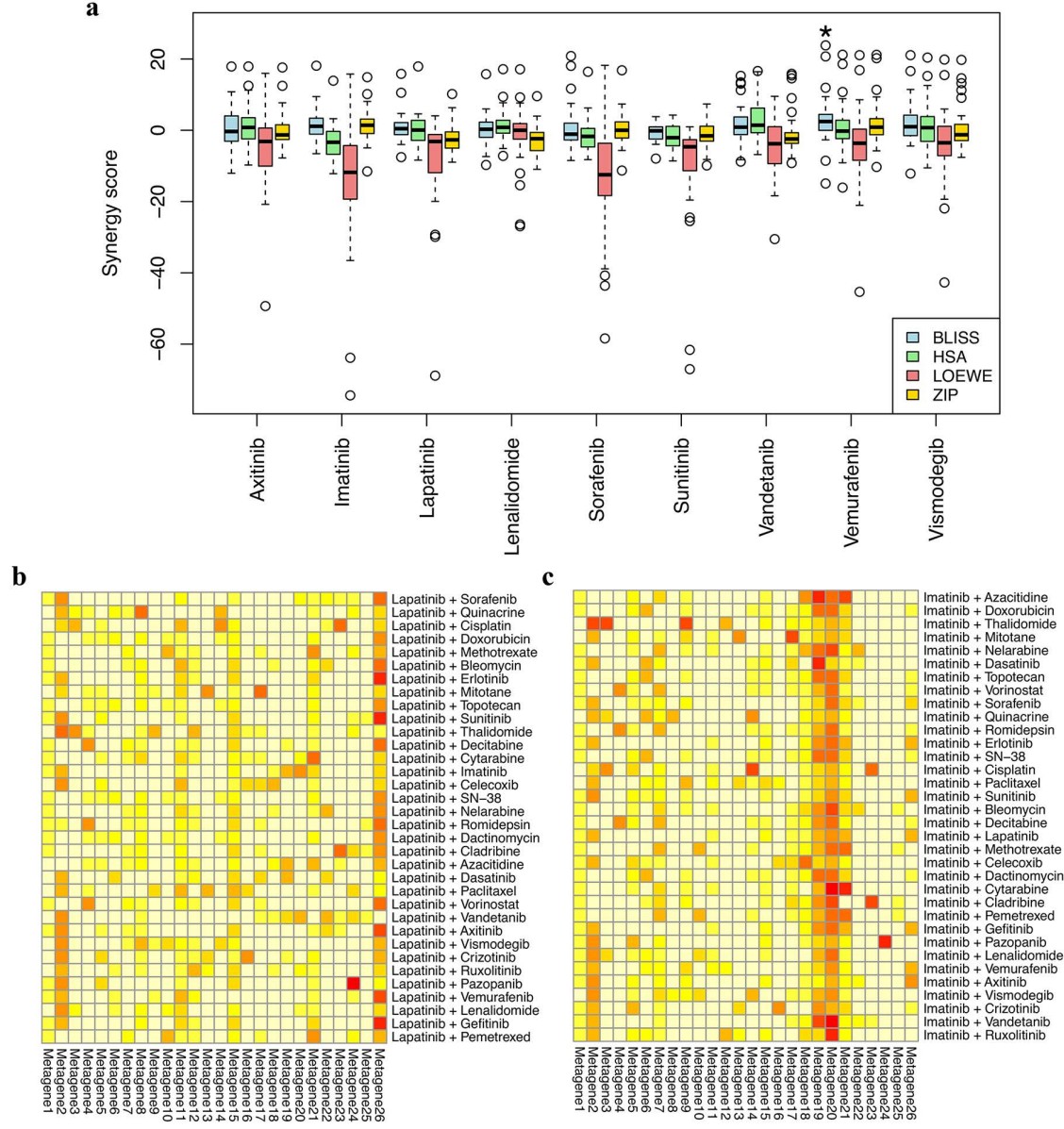

**Fig 7. Synergy scores and metagene profiles of partner drugs. (a)** Synergy scores of nine partner drugs across all tested drug pairs. The asterisk (*) denotes that the Bliss scores of drug combinations involving vemurafenib are the only condition with synergy scores significantly higher than zero (one-sided one-sample Wilcoxon signed-rank test, FDR-adjusted p-value < 0.05). **(b-c)** Metagene profiles of drug combinations involving lapatinib and imatinib in the dataset.

MAPK signaling pathways. Below, we discuss evidence supporting the synergistic influence and the therapeutic potential of regulating these pathways.

## Co-targeting SRC and its compensatory pathways overcomes resistance to dasatinib

Dasatinib is a multi-target kinase inhibitor targeting the Src family kinases (SFKs), including Src, Fyn, Lyn, and Yes, which are frequently overexpressed or hyperactivated in KRAS-mutant cancers, including the A549 cell line [42]. SFKs are

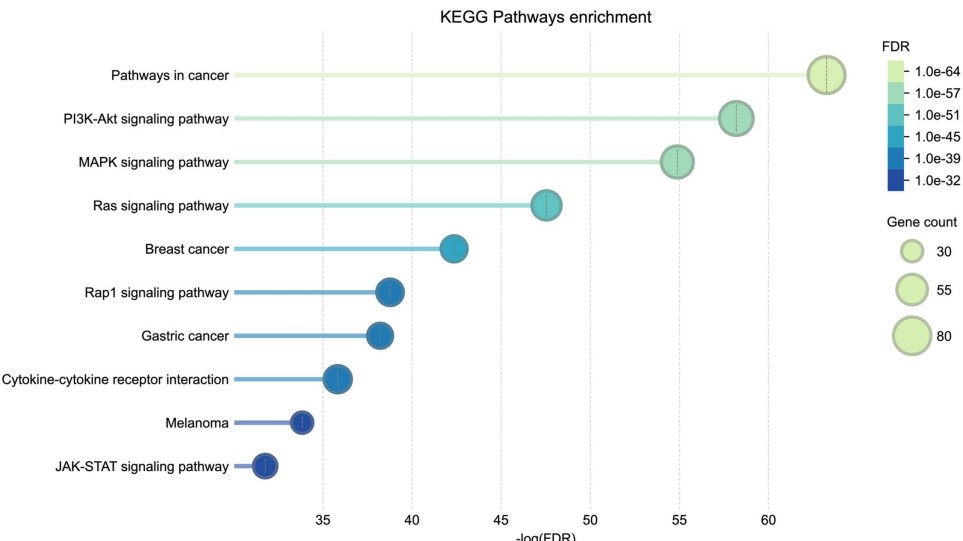

**Fig 8. KEGG pathway enrichment analysis of the top 200 genes in Metagene 2.** The figure was produced from the STRING database (version 12.0) [41], based on data from the Kyoto Encyclopedia of Genes and Genomes (KEGG) database [40].

significant activators of the PI3K/AKT pathway by phosphorylating the regulatory subunit (p85) of PI3K [43]. Additionally, SFKs are important activators of the RAS/MAPK pathway, primarily by phosphorylating adaptor proteins such as Shc, which then recruit GRB2/SOS to activate Ras [44]. This dual ability enables SFKs to promote both key proliferative and survival signals within the cell.

Although dasatinib effectively inhibits SFKs, A549 cells can bypass SFK inhibition through compensatory mechanisms. For example, EGFR can also recruit GRB2 and SOS, which ultimately activate RAS [45]. Activated receptor tyrosine kinases (RTKs), such as EGFR, HER2, IGF1R, VEGFR, and PDGFR, stimulate both the RAS/RAF/MEK/ERK and PI3K/AKT/mTOR pathways [46]. This independent mechanism highlights the versatility and complexity of RTK signaling networks. Our network propagation analysis reveals that synergy with dasatinib can be achieved by co-targeting these compensatory proteins that regulate the interconnected RAS/MAPK and PI3K/AKT signaling cascades. For example, the combination of dasatinib and axitinib (FLT1 and FLT4 inhibitor) results in HSA, ZIP, and Bliss scores of 11.1, 7.7, and 9.4, respectively. Dasatinib combined with vandetanib, a multi-RTK inhibitor, produces a synergy score above 13 for HSA, ZIP, and Bliss.

### Inhibiting the PI3K/AKT pathway reduces Tau activity and sensitizes cancer cells to paclitaxel

Paclitaxel is widely used in cancer treatment and often relies on combination therapies to enhance its effectiveness, as its efficacy is influenced by cell signaling pathways like PI3K/AKT [47]. One of paclitaxel's anticancer actions is stabilizing microtubules. Normally, microtubules undergo balanced assembly and disassembly through α/β-tubulin heterodimers and GTP hydrolysis. Paclitaxel binds to the N-terminus of the β-tubulin subunit, stabilizing microtubules and blocking their depolymerization. This stabilization disrupts normal cell cycle progression, leading to apoptosis [48].

In our network, paclitaxel interacts with MAPT (Tau protein), which also binds microtubules to stabilize them. Tau expression correlates with responses to microtubule-targeting agents like paclitaxel and other cancer therapies, highlighting its potential influence on cancer outcomes [49]. Tau binds to β-tubulin at both the outer and inner surfaces of microtubules, occupying the same binding site as paclitaxel and thereby competing with the drug [50]. Tau stabilizes microtubules in a manner similar to paclitaxel, but with a lower affinity and greater reversibility [51].

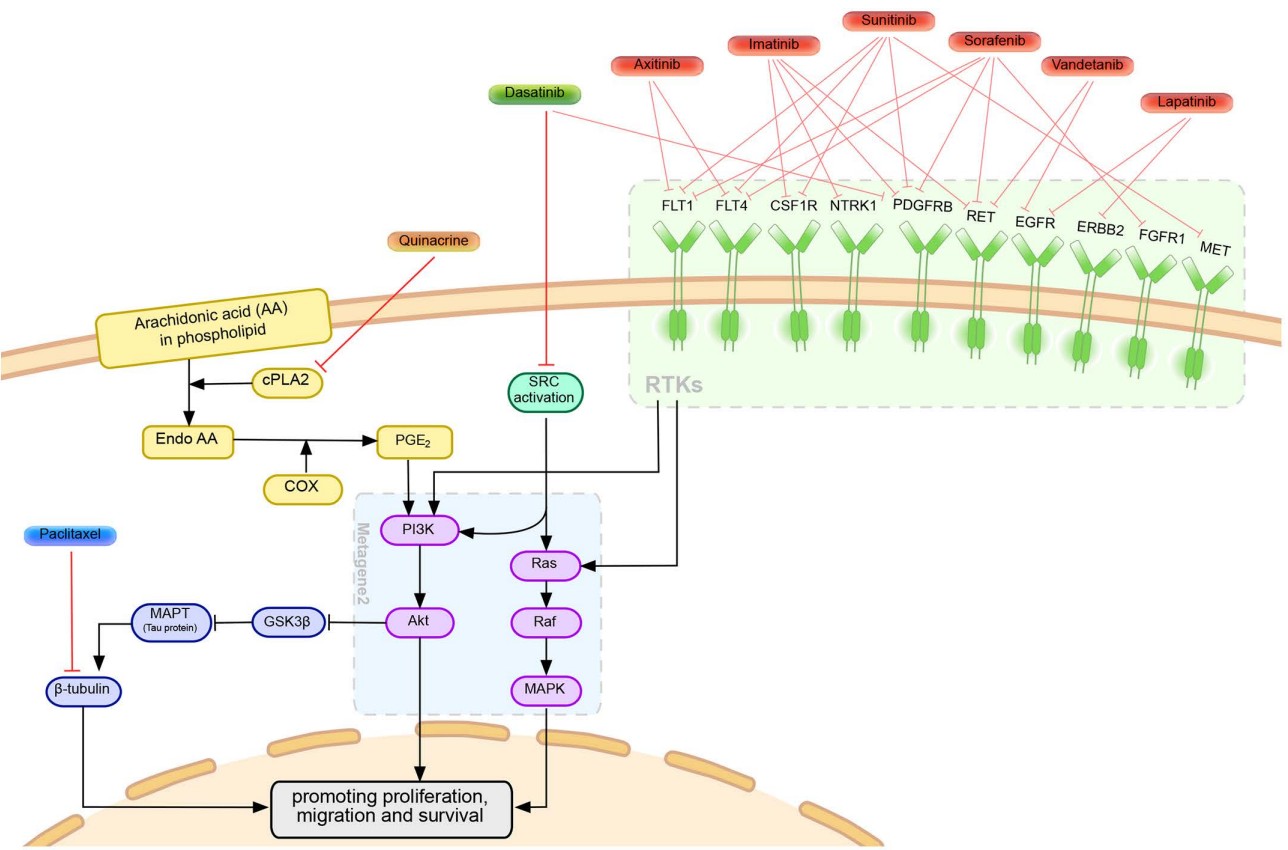

**Fig 9. Mechanisms of highly synergistic drug combinations in A549 by simultaneously regulating the interconnected RAS/MAPK and PI3K/AKT signaling pathways.**

Studies have shown that inhibiting the PI3K/AKT pathway can enhance sensitivity to microtubule-targeting drugs like docetaxel [52]. Specifically, the PI3K/AKT/GSK-3β signaling pathways are directly linked to Tau. Activated AKT phosphorylates and inactivates GSK-3β, which in turn reduces phosphorylation of Tau. This dephosphorylated Tau is highly functional in binding to and stabilizing microtubules. Since Tau proteins compete with paclitaxel for binding, an overactive PI3K/AKT pathway consequently reduces cellular sensitivity to paclitaxel. Our analysis consistently reveals that drug combinations enhancing paclitaxel's synergistic effect show an enriched score on Metagene 2, which is involved in the PI3K/AKT pathway.

In addition, research indicates that HGF inhibitors, which inhibit PI3K and AKT phosphorylation, increase apoptosis induced by paclitaxel in A549 cells [53]. This suggests that inhibiting HGF and its downstream PI3K/AKT signaling may be a potential therapeutic approach for overcoming paclitaxel resistance. Our analysis supports this finding, demonstrating that drugs that disrupt the PI3K pathway exhibit strong synergy when used in combination with paclitaxel. For example, lapatinib's mechanism of action involves inhibiting the upstream activation of HER2 (ERBB2) and EGFR, thereby suppressing the PI3K/AKT/mTOR pathway. Lapatinib in combination with paclitaxel has the HSA, ZIP, and Bliss scores of 17.90, 10.17, and 15.83, respectively.

## Disrupting the PI3K/AKT and arachidonic acid pathways enhances anticancer efficacy

Quinacrine, also known as mepacrine, was first introduced as an antimalarial drug but has more recently gained attention for its anticancer properties [54]. According to DrugBank, quinacrine inhibits calcium-independent phospholipase A2

(PLA2G6) and cytosolic phospholipase A2 (PLA2G4A), which participate in the arachidonic acid pathway. Down-regulating this pathway has demonstrated chemopreventive potential in several cancers, including lung cancer [55–58]. This pathway is initiated by the activation of phospholipase A2 (PLA2), which releases arachidonic acid from the cell membrane. This intracellular arachidonic acid serves as the substrate for cyclooxygenase enzymes, including cyclooxygenase-2 (COX-2), which converts it into various prostaglandins, including prostaglandin $E_2$ ($PGE_2$). Prostaglandins, in turn, act as important signaling molecules in inflammation, cell growth, survival, and invasion, fostering tumorigenesis across various carcinomas.

$PGE_2$ has been shown to promote survival in lung adenocarcinoma cell lines, including A549, by activating the PI3K/AKT signaling cascade [59]. On the other hand, the PI3K/AKT signaling pathway is proposed to promote COX-2 expression via CREB, thereby forming a positive feedback loop between the PI3K/AKT and arachidonic acid pathways [59]. The combination of inhibitors for COX-2 and AKT has been shown to have a synergistic effect on the proliferation and migration of multiple lung adenocarcinoma cell lines [59].

Consistent with this evidence, quinacrine (an inhibitor of the arachidonic acid pathway) shows synergistic activity with many tyrosine kinase inhibitors that act upstream of the PI3K/AKT pathway. For instance, quinacrine shows the HSA, ZIP, and Bliss synergy scores above 15 with axitinib (FLT1 and FLT4 inhibitor), crizotinib (ALK and MET inhibitor), and vandetanib (EGFR and RET inhibitor). Our analysis also identified the combinations of celecoxib (a COX-2 inhibitor) with the tyrosine kinase inhibitors crizotinib and pazopanib as highly synergistic (HSA and Bliss scores consistently above 6 and 9, respectively).

### Other synergistic combinations strongly signal Metagene 2, enabling simultaneous inhibition of PI3K/AKT or RAS/MAPK and compensatory pathways

Additionally, within the seven identified clusters, we found several other interesting, highly synergistic drug combinations that exhibit a strong Metagene 2 signal (Fig. 5). For example, Selumetinib (MAP2K1 inhibitor) + AZ13397262 (ALK inhibitor) simultaneously inhibits the MAPK and RTK pathways. Pictilisib (PIK3CG inhibitor) + AZ13397262 (ALK inhibitor), and Pictilisib (PIK3CG inhibitor) + HG-14-10-04 (ALK/EGFR inhibitor) facilitate the simultaneous inhibition of the PI3K and RTK pathways.

We also found that vismodegib, a targeted inhibitor of Smoothened (SMO), a key component of the Hedgehog signaling pathway, exhibits strong synergistic interactions with both dasatinib and quinacrine, as evidenced by notable synergy scores across HSA, ZIP, and Bliss models. There is extensive evidence of crosstalk between the hedgehog and PI3K/AKT or RAS/MAPK pathways [60], suggesting that the ability of hedgehog signaling to drive cell survival and drug resistance may contribute to the enhanced synergistic effects observed when vismodegib is combined with agents such as dasatinib and quinacrine.

## Discussion

The challenge of drug resistance in non-small cell lung cancer (NSCLC) necessitates a shift from single-target therapeutics to rational, multi-target combination strategies. The underlying rationale for this shift stems from observations that combining interventions on functionally proximal genes leads to greater efficacy than using single agents [9,25]. Our computational framework, which integrates Random Walk with Restart (RWR) and Graph-regularized Non-negative Matrix Factorization (GNMF), is explicitly designed to resonate with the rationale of co-targeting functionally proximal genes by bridging the gap between drug targets and complex cellular signaling cascades. Specifically, our approach reduces the complex 2100-gene network profiles into 26 functional metagenes—groups of genes that share functionally proximal relationships, shaped by the topology of the cancer network.

The use of metagenes provides a higher level of biological meaning and interpretation than the individual gene propagation scores. By clustering drug combinations based on their metagene scores in matrix *H,* we are essentially grouping

drug combinations that similarly affect the same biological processes. This shift to pathway-level insights provides a more detailed understanding of drug synergy, enabling the successful identification and biological interpretation of synergistic mechanisms shared across drug combination clusters.

A central finding of this study is the consistent role of Metagene 2 in driving synergy across seven of the ten highly enriched synergistic clusters. Pathway enrichment analysis conclusively linked Metagene 2 to the highly interconnected survival and proliferation signaling hubs: RAS, MAPK, and PI3K/AKT pathways. This strongly suggests that effective combination therapy in A549 requires simultaneous disruption of this integrated network, rather than targeting individual nodes or pathways in isolation.

Our approach validates the principle that synergy is achieved through the coordinated inhibition of compensatory or bypass pathways in the cell's signaling network. Pioneering network-based approaches, such as those by Cheng et al. (2019) [25], have successfully utilized topological proximity to predict synergy, providing valuable insights into drug interactions based on the distance between drug targets and disease modules. Building on this foundation, our study employs Random Walk with Restart (RWR) to complement these proximity metrics by capturing the network-wide propagation of drug effects, revealing broader systemic impacts that local distance measures may overlook. Furthermore, a distinct innovation of our work is the adaptation of Graph-regularized Non-negative Matrix Factorization (GNMF) to the analysis of drug combinations. This approach identifies 'metagene' profiles, allowing distinct drug pairs to be grouped by their shared mechanism of action.

Our analysis demonstrated that high synergy and strong Metagene 2 activity were not inherent properties of any single drug, but arose specifically from the drug-drug interaction within the combination. This emphasizes the utility of our RWR-GNMF method in capturing these non-linear, system-level effects that traditional single-target analyses often miss. Furthermore, the ability of Metagene 2 to clearly differentiate between high- and low-synergy groups for key drugs like dasatinib, paclitaxel, and quinacrine confirms its power as a predictive and explanatory biomarker for combination efficacy.

Importantly, the framework is adaptable to other NSCLC subtypes and pan-cancer analyses, extending beyond the A549 lung adenocarcinoma cell line. The core workflow involves constructing a context-specific network by integrating gene mutation status from the cell line of interest. Therefore, extending this to other subtypes requires tailoring the generic network using mutation or transcriptomic profiles specific to those cell lines. Furthermore, the framework can facilitate pan-cancer analysis by aggregating propagated profiles from diverse cell lines into a single matrix for factorization. This approach will enable the identification of universal metagenes that represent fundamental synergy mechanisms conserved across cancer types, as well as metagenes unique to specific cancer subtypes.

A key limitation of the current framework is its restriction to drugs with inhibitory mechanisms. This choice was primarily motivated by the fact that most drugs retrieved from the database are inhibitors, reflecting the dominant therapeutic strategy of suppressing oncogenic signaling pathways. Focusing exclusively on inhibitory effects enables a clear, unidirectional interpretation of signal propagation within the cancer network, as drug action is consistently modeled as the suppression of protein activity. In contrast, including activators or stabilizers would introduce conflicting signal directions, which greatly increases analytical complexity. In addition, two drugs that share the same protein targets but one is an activator, and the other an inhibitor, would have the same propagation profile, which complicates interpretation. We recognize that this exclusion represents a limitation of the present approach. Future directions should address this by modeling bidirectional or context-dependent signaling to capture the interplay between pathway agonists and antagonists.

While our metagene-based framework provides valuable mechanistic insight, its effectiveness relies on the quality of the network and the drug-target data. To improve the work, it is recommended to expand the network nodes and interactions to include a broader range of biological pathways. It is also important to carefully evaluate the quality of the network to ensure that the connections are accurate and reflect true biological relationships. Furthermore, increasing the number of drug combinations in the dataset will provide a more comprehensive basis for analysis, allowing for better identification of potential synergies and enhancing the overall reliability of the findings.

 

## Conclusions

We developed a computational framework that utilizes Random Walk with Restart (RWR) and Graph-regularized Non-negative Matrix Factorization (GNMF) to help identify and understand synergistic drug combinations in the A549 cell line. Through our analysis, we discovered a key factor—referred to as Metagene 2—that is a driving force behind the effectiveness of these drug combinations. This metagene represents a crucial survival network formed by the interconnected RAS/MAPK and PI3K/AKT signaling pathways. The consistently strong signal of Metagene 2 in synergistic drug pairs indicates that their combined effectiveness hinges on regulating this central signaling network. Overall, our metagene-based classification framework not only provides a clearer understanding of the biological mechanisms behind drug synergy but also offers a valuable resource for designing targeted, multi-drug therapies for NSCLC.

## Supporting information

**S1 Text. Supporting information text.**
(DOCX)

**S1 File. The dataset of 607 drug combinations used in the analysis.**
(CSV)

**S2 File. 45 unique drugs involved in the dataset and their targets.**
(CSV)

**S3 File. The interaction pairs in the cancer network, including 2100 genes and 162,211 interactions.**
(CSV)

**S4 File. The propagated profiles of 607 drug combinations across 2100 genes obtained after network propagation.**
(CSV)

## Acknowledgments

T.L. acknowledges the use of Grammarly to correct grammar and improve language clarity. The authors appreciate the editor and reviewers' valuable suggestions, which have greatly improved the manuscript.

## Author contributions

**Conceptualization:** Teerasit Termsaithong, Teeraphan Laomettachit.

**Formal analysis:** Comkrit Lomloy, Piyanut Ratphibun Yamashita, Teerasit Termsaithong, Teeraphan Laomettachit.

**Funding acquisition:** Teerasit Termsaithong, Teeraphan Laomettachit.

**Investigation:** Comkrit Lomloy, Piyanut Ratphibun Yamashita, Teerasit Termsaithong, Teeraphan Laomettachit.

**Methodology:** Comkrit Lomloy, Teerasit Termsaithong, Teeraphan Laomettachit.

**Supervision:** Teerasit Termsaithong, Teeraphan Laomettachit.

**Visualization:** Comkrit Lomloy, Teeraphan Laomettachit.

**Writing – original draft:** Comkrit Lomloy, Teeraphan Laomettachit.

**Writing – review & editing:** Comkrit Lomloy, Piyanut Ratphibun Yamashita, Teerasit Termsaithong, Teeraphan Laomettachit.

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
