## [Decision Letter · Decision Letter 0]

12 Dec 2025

PONE-D-25-58836Uncovering the mechanisms of synergistic drug combinations in non-small cell lung cancer through metagene-based classificationPLOS One

Dear Dr. Laomettachit,

Thank you for submitting your manuscript to PLOS ONE. After careful consideration, we feel that it has merit but does not fully meet PLOS ONE’s publication criteria as it currently stands. Therefore, we invite you to submit a revised version of the manuscript that addresses the points raised during the review process.

We look forward to receiving your revised manuscript.

Kind regards,

Erfan Ghadirzadeh, MD

Academic Editor

PLOS One

Journal Requirements:

4. We notice that your supplementary figures are uploaded with the file type 'Figure'. Please amend the file type to 'Supporting Information'. Please ensure that each Supporting Information file has a legend listed in the manuscript after the references list.

Reviewer's Responses to Questions

**Comments to the Author**

1. Is the manuscript technically sound, and do the data support the conclusions?

Reviewer #1: Yes

Reviewer #2: Yes

Reviewer #3: Yes

2. Has the statistical analysis been performed appropriately and rigorously? 

Reviewer #1: Yes

Reviewer #2: Yes

Reviewer #3: Yes

3. Have the authors made all data underlying the findings in their manuscript fully available?

Reviewer #1: Yes

Reviewer #2: Yes

Reviewer #3: Yes

4. Is the manuscript presented in an intelligible fashion and written in standard English?

Reviewer #1: Yes

Reviewer #2: Yes

Reviewer #3: Yes

5. Review Comments to the Author

Reviewer #1: Title: “Uncovering the mechanisms of synergistic drug combinations in non-small cell lung cancer through metagene-based classification”

The study addresses the synergistic drug combinations that simultaneously target multiple signaling pathways to overcome anticancer drug resistance. The authors identified the metagene 2 as a driving force behind the effectiveness of the drug combinations by the interconnected RAS/MAPK and PI3K/AKT signaling pathways. This metagene-based framework provides a valuable resource for designing targeted multi-drug therapies for NSCLC.

The subject is interesting and addresses the important issue of drug resistance in cancer therapy. The research design is sound, well conducted, and the manuscript is well-written and may be suitable for publication.

Reviewer #2: In the current manuscript, the authors present methodologically, systems-level framework for understanding mechanistic drivers of drug synergy in the A549 Non-small cell lung cancer (NSCLC) cell line. By combining frameworks, Random Walk with Restart (RWR) and Graph-regularized Non-negative Matrix Factorization (GNMF), the authors derive pathway-level “metagene” signatures which enable interpretable drug combinations and mechanistic hypothesis generation. In previous studies, Cheng et al. demonstrated that analyzing drug–disease and drug–drug proximities within the human interactome can successfully predict synergistic combinations by exploiting the spatial geometry of disease modules (doi.org/10.1038/s41467-019-09186-x). Another study showed that propagating somatic mutation profiles over a protein–protein interaction network, followed by matrix factorization, reveals biologically meaningful tumor subtypes predictive of clinical outcomes (doi.org/10.1038/nmeth.2651; doi.org/10.1093/bioinformatics/btaa1099). The present manuscript is conceptually consistent with these approaches but extends them through a more expressive metagene representation. This extension produces pathway-level signatures rather than topological proximity scores, enabling the mechanistic insight stated by the authors. The methods presented are well aligned with previous network-based stratification (NBS). The current study applies the same conceptual machinery to drug combination profiles, using an A549-specific cancer network and graph-regularized NMF to distill 2100-gene propagated vectors into 26 biologically coherent metagenes. Here, by identifying a dominant metagene (Metagene 2) whose constituent genes map to PI3K/AKT, RAS, and MAPK signaling, the authors provide a direct mechanistic rationale for why certain drug pairs (e.g., dasatinib–axitinib, lapatinib–paclitaxel, quinacrine–TKI combinations) exhibit strong synergy. This pathway-level interpretability fills an important gap in current drug-combination research and is within the scope of the journal. The manuscript can be improved based on the comments below:

1. Although the authors note this in the Discussion, elaborating on how the framework might extend to other NSCLC subtypes or pan-cancer analyses would add depth to the conclusion.

2. Authors should provide clear justification for restricting analysis to drugs with inhibitory targets only and discuss potential implications for activators or stabilizers.

3. Please include FDR-adjusted p-values directly in Figs. 2 to 4, 6c and 7a.

Reviewer #3: 1- The introduction part is lengthy and and be trimmed.

2- The discussion part should be strengthen and the authors should highlight the new aspects of the current study, compare this with the similar manuscript regarding the issue.

6. PLOS authors have the option to publish the peer review history of their article (what does this mean?). If published, this will include your full peer review and any attached files.

Reviewer #1: No

Reviewer #2: No

Reviewer #3: No

---

## [Author Response · Author response to Decision Letter 1]

2 Jan 2026

Please find our detailed point-by-point responses to the reviewers' comments in the attached file labeled 'Response to Reviewers'.

---

## [Decision Letter · Decision Letter 1]

13 Feb 2026

Uncovering the mechanisms of synergistic drug combinations in non-small cell lung cancer through metagene-based classification

PONE-D-25-58836R1

Dear Dr. Laomettachit,

We’re pleased to inform you that your manuscript has been judged scientifically suitable for publication and will be formally accepted for publication once it meets all outstanding technical requirements.

Kind regards,

Erfan Ghadirzadeh, MD

Academic Editor

PLOS One

Additional Editor Comments (optional):

Reviewers' comments:

Reviewer's Responses to Questions

**Comments to the Author**

1. If the authors have adequately addressed your comments raised in a previous round of review and you feel that this manuscript is now acceptable for publication, you may indicate that here to bypass the “Comments to the Author” section, enter your conflict of interest statement in the “Confidential to Editor” section, and submit your "Accept" recommendation.

Reviewer #1: All comments have been addressed

Reviewer #2: All comments have been addressed

2. Is the manuscript technically sound, and do the data support the conclusions?

Reviewer #1: Yes

Reviewer #2: Yes

3. Has the statistical analysis been performed appropriately and rigorously? 

Reviewer #1: Yes

Reviewer #2: Yes

4. Have the authors made all data underlying the findings in their manuscript fully available?

Reviewer #1: Yes

Reviewer #2: Yes

5. Is the manuscript presented in an intelligible fashion and written in standard English?

Reviewer #1: Yes

Reviewer #2: Yes

6. Review Comments to the Author

Reviewer #1: The manuscript was improved after addressing the reviewer suggestions and become suitable for publication

Reviewer #2: The authors have satisfactorily answered all questions raised. The modifications made to the manuscript and additional data support the conclusions of the study.

7. PLOS authors have the option to publish the peer review history of their article (what does this mean?). If published, this will include your full peer review and any attached files.

Reviewer #1: No

Reviewer #2: No

---

## [Editor Report · Acceptance letter]

PONE-D-25-58836R1

PLOS One

Dear Dr. Laomettachit,

I'm pleased to inform you that your manuscript has been deemed suitable for publication in PLOS One. Congratulations! Your manuscript is now being handed over to our production team.

Kind regards,

on behalf of

Dr. Erfan Ghadirzadeh

Academic Editor

PLOS One